Journal of
open psychology data

# The Replication Database: Documenting the Replicability of Psychological Science

DATA PAPER

**LUKAS RÖSELER** (ID)

**LEONARD KAISER**

**CHRISTOPHER DOETSCH**

**NOAH KLETT**

**CHRISTIAN SEIDA** (ID)

**ASTRID SCHÜTZ** (ID)

**BALAZS ACZEL** (ID)

**NADIA ADELINA** (ID)

**VALERIA AGOSTINI** (ID)

**SAMUEL ALARIE**

**NIHAN ALBAYRAK-AYDEMIR** (ID)

**ALAA ALDOH** (ID)

**ALI H. AL-HOORIE** (ID)

**FLAVIO AZEVEDO** (ID)

**BRADLEY J. BAKER** (ID)

**CHARLOTTE LILIAN BARTH**

**JULIA BEITNER** (ID)

**CAMERON BRICK** (ID)

**HILMAR BROHMER** (ID)

**SUBRAMANYA PRASAD CHANDRASHEKAR** (ID)

**KAI LI CHUNG** (ID)

**JAMIE P. COCKCROFT** (ID)

**JAMIE CUMMINS** (ID)

**VERONICA DIVEICA** (ID)

**TSVETOMIRA DUMBALSKA** (ID)

**EMIR EFENDIC** (ID)

**MAHMOUD ELSHERIF** (ID)

**THOMAS EVANS** (ID)

**GILAD FELDMAN** (ID)

**ADRIEN FILLON** (ID)

**NICO FÖRSTER** (ID)

**JORIS FRESE** (ID)

**OLIVER GENSCHOW** (ID)

**VAITSA GIANNOULI** (ID)

**BILJANA GJONESKA** (ID)

**TIMO GNAMBS** (ID)

**AMÉLIE GOURDON-KANHUKAMWE** (ID)

**CHRISTOPHER J. GRAHAM** (ID)

**HELENA HARTMANN** (ID)

**CLOVE HAVIVA** (ID)

**ALINA HERDERICH** (ID)

**LEON P. HILBERT** (ID)

**DARÍAS HOLGADO** (ID)

**IAN HUSSEY** (ID)

**ZLATOMIRA G. ILCHOVSKA** (ID)

**TAMARA KALANDADZE** (ID)

**VELI-MATTI KARHULAHTI** (ID)

**LEON KASSECKERT**

**MAREN KLINGELHÖFER-JENS** (ID)

**ALINA KOPPOLD** (ID)

**MAX KORBMACHER** (ID)

**LOUISA KULKE** (ID)

**NICLAS KUPER** (ID)

**ANNALISE LAPLUME** (ID)

**GAVIN LEECH** (ID)

**FELINE LOHKAMP**

**NIGEL MANTOU LOU** (ID)

**DERMOT LYNOTT** (ID)

**MAXIMILIAN MAIER** (ID)

]u[ ubiquity press

**CORRESPONDING AUTHOR:**
**Lukas Röseler**

Münster Center for Open Science, University of Münster, Germany; Institute of Psychology, University of Bamberg, Germany

Lukas.roeseler@uni-muenster.de

**KEYWORDS:**
Replication; replication crisis; database; open science; collaborative; credibility revolution; meta science

**TO CITE THIS ARTICLE:**
Röseler, L., Kaiser, L., Doetsch, C., Klett, N., Seida, C., Schütz, A., Aczel, B., Adelina, N., Agostini, V., Alarie, S., Albayrak-Aydemir, N., Aldoh, A., Al-Hoorie, A. H., Azevedo, F., Baker, B. J., Barth, C. L., Beitner, J., Brick, C., Brohmer, H., Chandrashekar, S. P., Chung, K. L., Cockcroft, J. P., Cummins, J., Diveica, V., Dumbalska, T., Efendic, E., Elsherif, M., Evans, T., Feldman, G., Fillon, A., Förster, N., Frese, J., Genschow, O., Giannouli, V., Gjoneska, B., Gnambs, T., Gourdon-Kanhukamwe, A., Graham, C. J., Hartmann, H., Haviva, C., Herderich, A., Hilbert, L. P., Holgado, D., Hussey, I., Ilchovska, Z. G., Kalandadze, T., Karhulahti, V.-M., Kasseckert, L., Klingelhöfer-Jens, M., Koppold,

**MARIA MEIER** (iD)

**MARIA MONTEFINESE** (iD)

**DAVID MOREAU** (iD)

**KELLEN MRKVA** (iD)

**MONIKA NEMCOVA** (iD)

**DANNA OOMEN** (iD)

**JULIAN PACKHEISER** (iD)

**SHUBHAM PANDEY**

**FRANK PAPENMEIER** (iD)

**MARIOLA PARUZEL-CZACHURA** (iD)

**YURI G. PAVLOV** (iD)

**ZORAN PAVLOVIĆ** (iD)

**CHARLOTTE R. PENNINGTON** (iD)

**MERLE-MARIE PITTELKOW** (iD)

**WILLEMIJN PLOMP** (iD)

**PAUL E. PLONSKI** (iD)

**EKATERINA PRONIZIUS** (iD)

**ANDREW ADRIAN PUA** (iD)

**KATARZYNA PYPNO-BLAJDA** (iD)

**MANUEL RAUSCH** (iD)

**TOBIAS R. REBHOLZ** (iD)

**ELENA RICHERT** (iD)

**JAN PHILIPP RÖER** (iD)

**ROBERT ROSS** (iD)

**KATHLEEN SCHMIDT** (iD)

**ALEKSANDRINA SKVORTSOVA** (iD)

**MATTHIAS F. J. SPERL** (iD)

**ALVIN W. M. TAN** (iD)

**J. LUKAS THÜRMER** (iD)

**ALEKSANDRA TOŁOPIŁO** (iD)

**WOLF VANPAEMEL** (iD)

**LEIGH ANN VAUGHN** (iD)

**STEVEN VERHEYEN** (iD)

**LUKAS WALLRICH** (iD)

**LUCIA WEBER**

**JULIA K. WOLSKA** (iD)

**MIRELA ZANEVA** (iD)

**YIKANG ZHANG** (iD)

*Author affiliations can be found in the back matter of this article

A., Korbmacher, M., Kulke, L., Kuper, N., LaPlume, A., Leech, G., Lohkamp, F., Lou, N. M., Lynott, D., Maier, M., Meier, M., Montefinese, M., Moreau, D., Mrkva, K., Nemcova, M., Oomen, D., Packheiser, J., Pandey, S., Papenmeier, F., Paruzel-Czachura, M., Pavlov, Y. G., Pavlović, Z., Pennington, C. R., Pittelkow, M.-M., Plomp, W., Plonski, P. E., Pronizius, E., Pua, A. A., Pypno-Blajda, K., Rausch, M., Rebholz, T. R., Richert, E., Röer, J. P., Ross, R., Schmidt, K., Skvortsova, A., Sperl, M. F. J., Tan, A. W. M., Thürmer, J. L., Tołopiło, A., Vanpaemel, W., Vaughn, L. A., Verheyen, S., Wallrich, L., Weber, L., Wolska, J. K., Zaneva, M., & Zhang, Y. (2024). The Replication Database: Documenting the Replicability of Psychological Science. *Journal of Open Psychology Data,* 12: 8, pp. 1–23. DOI: https://doi.org/10.5334/jopd.101

## ABSTRACT

In psychological science, replicability—repeating a study with a new sample achieving consistent results (Parsons et al., 2022)—is critical for affirming the validity of scientific findings. Despite its importance, replication efforts are few and far between in psychological science with many attempts failing to corroborate past findings. This scarcity, compounded by the difficulty in accessing replication data, jeopardizes the efficient allocation of research resources and impedes scientific advancement. Addressing this crucial gap, we present the *Replication Database* (https://forrt-replications.shinyapps.io/fred_explorer), a novel platform hosting 1,239 original findings paired with replication findings. The infrastructure of this database allows researchers to submit, access, and engage with replication findings. The database makes replications visible, easily findable via a graphical user interface, and tracks replication rates across various factors, such as publication year or journal. This will facilitate future efforts to evaluate the robustness of psychological research.

# (1) BACKGROUND

In scientific research, almost every new hypothesis is based on previous findings; this epistemic connectedness is a core feature of science (Hoyningen-Huene, 2013). Scientific replication – the process of retesting a hypothesis with new data to determine whether the original study's conclusions can be supported (Parsons et al., 2022)–is essential for building a robust body of knowledge and ensuring the integrity and reliability of scientific research. From a theory-driven perspective, if the findings on which a theory has been built cannot be replicated, the theory needs to be discarded or modified. From a phenomenon-driven perspective, replication failures can shed light on important confounding factors that need to be addressed for the phenomenon or "effect" to be detected (e.g., Calder, Phillips, & Tybout, 1981). From an efficiency standpoint, it is important to know which scientific findings are replicable to ensure optimal allocation of resources and strategic steering of future work. Finally, replicability is an important part of building a more coherent *body* of evidence capable of informing practice and policy as a way to test the generalizability of a theory or procedure, especially in the causal claim of the theory (Syed, 2023). This can be done by more rigorously testing the heterogeneity of an effect through replication (Bryan et al., 2021; Syed, 2023). Robustness of effects through replication is one way to increase the quality of evidence for policy making (Brown et al., 2014). As a consequence, a lack of emphasis on replication research or reduced visibility of replications can hinder scientific progress and contribute to unnecessary waste of resources.

In psychological sciences, replication attempts have historically been rare (Koole & Lakens, 2012; Makel, Plucker, & Hegarty, 2012), but they have gained much attention in recent years through large-scale replication projects (e.g., Open Science Collaboration, 2015). Such attempts have identified relatively low replication rates (<60%; Camerer et al., 2016; Klein et al., 2014; Klein et al., 2018; Open Science Collaboration, 2015) with few exceptions (Protzko et al., 2024 but see Bak-Coleman & Devezer, 2023 for a comment; Soto, 2019). These findings have motivated claims that the psychological sciences are suffering from a 'replication crisis' (Maxwell et al., 2015; Nelson et al., 2018; Schooler, 2014) and are now undergoing a 'credibility revolution' (Korbmacher et al., 2023; Vazire, 2018). Concerns about replicability have therefore grown over the last decade, and have also been echoed in other sciences (e.g., Errington et al., 2021; Nosek & Errington, 2017). These concerns have led to substantially large collaborative efforts to enhance the quality of psychological research (e.g., Ebersole et al., 2020; Morey et al., 2016; Moshontz et al., 2018) and the wider academic system (e.g., Davis et al., 2018; Eder & Frings, 2021; Frith, 2020; Pennington, 2024; Silverstein et al., 2024; Stengers, 2016; Stewart et al., 2022). Moreover, individual researchers and smaller groups of researchers have started engaging in more replication research (e.g., Soderberg et al., 2021; Visser et al., 2022; Pavlov et al., 2021). Despite the growing number of replication attempts in the literature, no comprehensive database like this exists so far. Therefore, there is a clearly defined need to systematically track which studies have been subject to replication attempts and the outcome of those attempts.

We propose that continually and transparently tracking replication attempts in an organized and systematic way can increase trust in science, promote the development of robust theory-driven research, and optimize the use of academic and institutional resources. For this tracking, we have created the *Replication Database*. Our database will provide researchers, educators, students, and practitioners with systematized and low-barrier open access to previous findings. Thereby, it will help reduce the waste of research resources, as the results of studies traditionally considered as "unsuccessful" are often not published and land in the metaphorical "file drawer" (e.g., Kulke & Rakoczy, 2018). By using a public and crowdsourced database for replications, researchers may further circumvent journal gatekeeping (Mynatt et al., 1977; Sterling, 1959). Moreover, a replication database could be used by researchers to monitor and evaluate meta-scientific factors that may affect replicability, contributing to both the theoretical development of metascience as a discipline and evidence-based reformations improving replication research and its evaluation. For example, this curated resource of replication attempts could be the first step in the development of standards and guidelines to determine when an effect or non-effect can be considered 'replicable', ensuring clearer, multidimensional, and more nuanced understanding and definitions when we talk about "failed" or "(not) replicated" effects.

Therefore, we aggregated, transformed, and expanded datasets from large-scale replication attempts (e.g., Open Science Collaboration, 2015), publicly available lists of replications (e.g., LeBel et al., 2018; CurateScience, https://web.archive.org/web/20220128104303mp_/https://curatescience.org/app/replications), and individual replications conducted by ourselves or other researchers, with the ultimate aim to create a comprehensive replication database. Although the inclusion criteria for the database are not limited to psychology, most of the existing entries are based on original studies published in psychology journals. The current report provides a snapshot of 1,239 replication findings entered into the database. However, the database is intended as a living resource, and we are committed to updating it regularly as more replications occur to unceasingly facilitate finding, publishing, teaching, monitoring, and analyzing replications.

Researchers can freely use the dataset and/or an interactive Shiny Application (https://forrt-replications. shinyapps.io/fred_explorer, see Figure 1) to search and analyze the data. In addition, the Replication Database provides a short guide on the best practices of understanding replications, discussing key topics around replicability, such as: *What is the overall replication rate? What features characterize successful replication attempts? What attributes are associated with original studies that are replicable? How do replication rates vary over time and across fields?* These could be used as additional introductory teaching and learning resources.

## (2) METHODS

### 2.1 STUDY DESIGN

#### Inclusion Criteria

Inclusion criteria for the Replication Database were chosen liberally *a priori*. According to Hüffmeier et al. (2016), every study that tests the same hypothesis as a previous study could be deemed a replication. In our case, we required studies to specify which original study they had planned to replicate. As for research areas, studies from all social, cognitive, and beharioval sciences as well as medicine can be entered and validated.

The liberal definition of what constitutes a replication leads to variance in the closeness of replication studies. For example, some may reuse the same instructions, items, and analysis code, while others "merely" test the same hypothesis with newly created materials, in another language, and with a different type of sample. To capture these differences, we included optional variables about the similarities between original and replication study. These stem mostly from the Replication Recipe (Brandt

et al., 2014). Apart from an open-ended variable where all differences can be explained and evaluated, specific variables let researchers indicate whether the closeness of instructions, measures, stimuli, etc. is "exact", "close", "different", whether it cannot be evaluated ("does not apply") or whether it is "unknown". Arguably, we cannot define for all possible cases whether changing the language of a validated questionnaire should be considered close, which is why we have to rely on contributors to make informed assessments and specify the differences in the open-ended question. We advise researchers using these variables, to let further people code the variables and assess inter-rater agreement.

Most replication studies feature a limited number of focal hypothesis tests that can be paired with tests from previous studies (e.g., two paired standardized effect sizes). The database structure allows for entering multiple results per sample so that results from structural equation models, functional magnetic resonance imaging (fMRI) data, or other types of data may also be entered (see also section "Database Structure"). For completeness, we also decided to include results from studies that cannot be converted to correlation coefficients (e.g., Cramer's V, Hazard Ratios, Bayes Factors). These cannot be included into meta-analyses or other kinds of quantitative summaries but are displayed when searching the database (e.g., via the reference list annotation tool). Finally, entries can optionally include test statistics, from which standardized effect sizes can be calculated.

### Database Structure

The dataset has a multilevel structure (see Figure 2). Each row represents one phenomenon or effect (e.g., "Facial redness increases perceived anger"), for which the original finding's reference, the replication study's

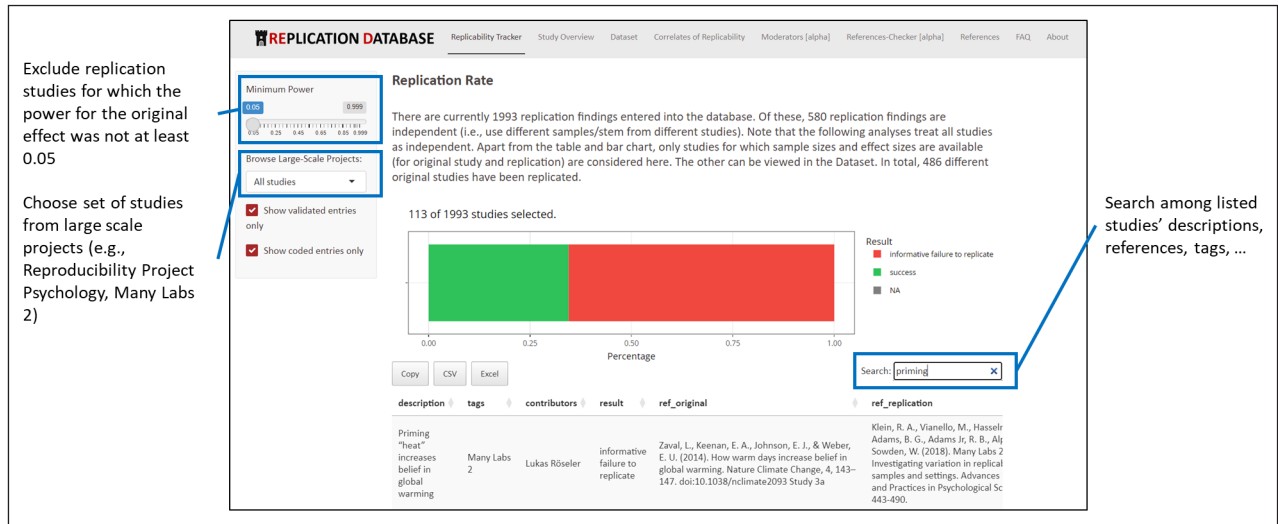

**Figure 1** Replication Tracker and example functions.

*Note.* Researchers can access the database to filter findings (e.g., for statistical power, validation status) and search among the entries. On the "Replicability Tracker" tab, replication rates for all selected findings are visualized. The high number of findings in the Figure is due to a more recent dataset included on the website.

reference, study numbers (when an article features multiple studies), standardized effect sizes, and sample sizes are coded. Additional metadata variables (e.g., differences between replication study and original study, journal that published the original study) are optionally coded.

In cases where a single replication study replicated an original effect in multiple ways (e.g., with several different items), we recommend documenting each effect separately for thoroughness, although this is not feasible for all projects (e.g., if results are only shared in an aggregated way as in Vaidis et al., 2024).

The database structure accommodates various complex scenarios such as multiple independent replications of the same original study, one single study that replicated multiple original studies, or one replication of two different original studies. Several frequent scenarios are discussed in detail below and depicted in Figure 2.

### One Single Study that Replicated One Original Study

In the least complicated case, there is one replication attempt entered into the database that corresponds to one original study. For example, Simmons and Nelson (2019) replicated Study 1b from Jami (2019) and reported the average effect size (effect sizes for all items separately are only visible in a plot). Thus, in the database, the average effect for each study is entered as one row.

If effect sizes for each of multiple items were coded, each pair of original and replication effect sizes would correspond to one row in the dataset and each row would be assigned the same values for the variable *id_sample*. If, for example, there is an entire correlation matrix for the pair of original and replication study, each pair of correlations will be entered in one row. Finally, if effect sizes for the original items plus a new item (i.e., an extension) are available, there can be five entries with the extension being coded as differing from the original study.

More complex studies may also nest replication effects of items or dependent variables in hypotheses (i.e., effect sizes are available for multiple dependent hypotheses and dependent variables). In the database, hypotheses and items can be specified in the "description" variable. As for collapsing or aggregating, coding was guided by what original effect sizes were available (e.g., ideally, every replication effect should be matched with an original effect).

### Multiple Independent Replications of the Same Original Study

Independence of tests can refer to samples consisting of different people or studies stemming from different laboratories. In the Replication Database, we refer to independence of samples. In the case of registered

replication reports (e.g., Bouwmeester et al., 2017), one original study is replicated by many different laboratories. In such a case, each laboratory's replication effect size is entered into the database with different values for the variable *id_sample*. The same pattern emerges if an effect is replicated by different laboratories. Note that for registered replication reports, it is also possible to "only" enter the aggregated replication effect size into the database (e.g., Vaidis et al., 2024 only shared the aggregate effect size in their report).

Note that the database entries' references are also supplemented by study number if more than one study is included in either report (e.g., "Cheung, B. Y., & Heine, S. J. (2015). The double-edged sword of genetic accounts of criminality: Causal attributions from genetic ascriptions affect legal decision making. Personality and Social Psychology Bulletin, 41(12), 1723–1738. *Study 3*" [emphasis added]). We plan to disentangle references and study numbers in the future (i.e., code them as two separate variables instead of one merged variable).

### One Single Study that Replicated Multiple Original Studies

Occasionally, data is collected in one study (or in other words, from one sample) and used to test multiple hypotheses. For example, Soto (2019) collected data from $N$ = 1,504 participants to compute 78 correlations for which previously published estimates had been available. In the Replication Database, these findings are represented as 78 rows that all have the same values for the variable *id_sample* and different original references, effect sizes, and descriptions.

### One Replication of Two Different Original Studies

If a replication report does not specify which original study it strives to replicate, the replication findings cannot be entered in the database. If, however, the replication is a replication of multiple original studies, several options arise: First, if for example, an original study has been replicated and now a second replication study is conducted, both replication studies are coded as replications of the original study. If, however, the first replication study introduces new features (e.g., the experimental manipulation has been altered) and the second replication study sticks with the alteration, it can be coded as a replication of the first replication. In a case, where a replication is a mix of two original studies (e.g., items from both original studies were mixed), the replication findings are entered twice (i.e., one time for each original study). This duplication can be identified via identical values in the variable *id_sample*. The upside of duplicating entries this way is that users of the database can find the replication via both of the original studies. Note that such cases are very rare.

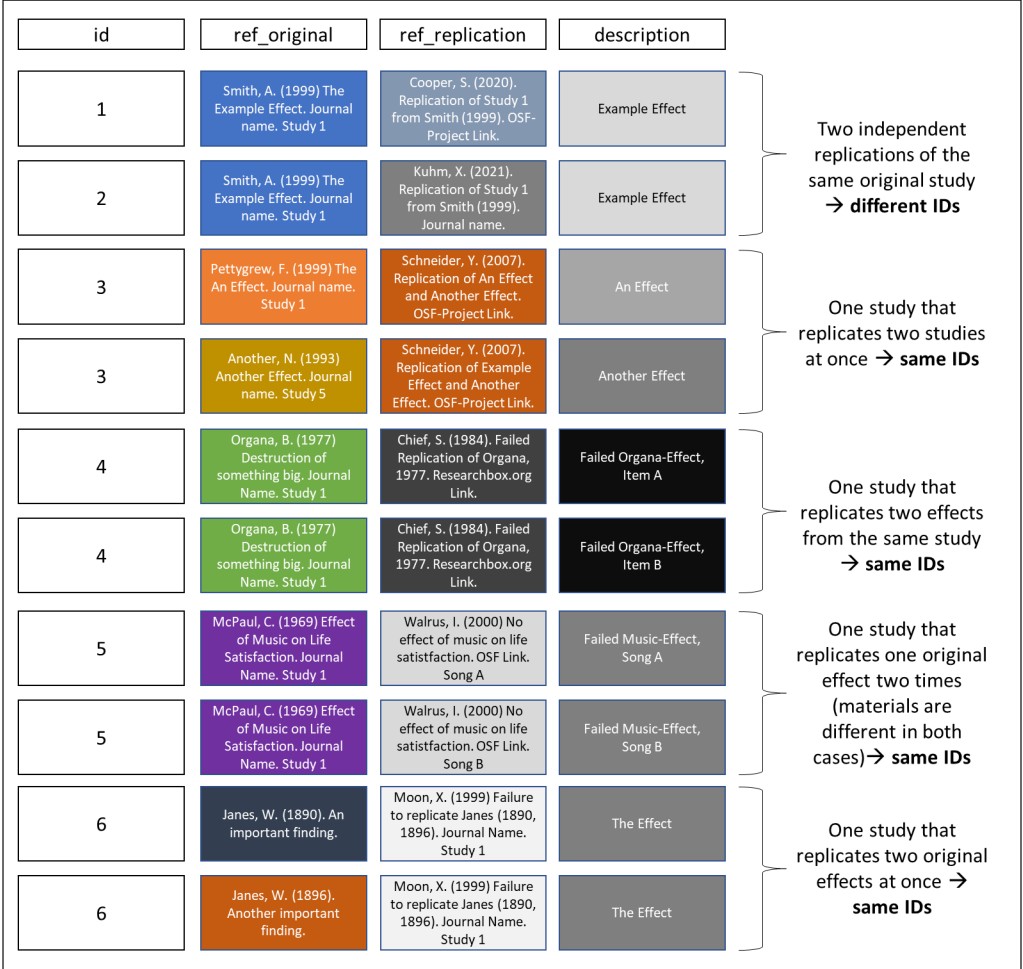

**Figure 2** Multilevel structure of the Replication database using fictitious data.

*Note.* OSF: Open Science Framework.

## Effect Size Conversion

The dataset includes effect sizes that were reported in the original and replication studies and – where possible – effect sizes converted to correlation coefficients to achieve commensurability. Effect sizes were converted to Pearson correlation coefficients using R (version 4.3.2; R Core Team, 2018) with the packages *esc* (Lüdecke, 2018), *metafor* (Viechtbauer, 2010), and *psychometric* (Fletcher, 2022). Data was further processed with: *dplyr* (Wickham et al., 2018), *lubridate* (Grolemund & Wickham, 2011), *pwr* (Champely, 2020), and *openxlsx* (Schauberger & Walker, 2021). The code to convert entries from the submission portal to the database structure (see section "Submission of Individual Entries") is freely available on the OSF at https://osf.io/2rv9z.

We kept the original effect sizes. In addition, we converted Odds Ratios, Cohen's *d*, η², *R*², and Cohen's *f* to correlation coefficients. φ coefficients were used as correlations without conversion (no conversion needed). Standardized regression coefficients, Cramer's *V*, Bayes Factors, Hazard Ratios, Cohen's *q*, Risk Ratios, Spearman's Rho, and Kendall's Tau were not converted and can thus not be included in meta-analysis of effect sizes (see Table 1).

| EFFECT SIZE OR TEST STATISTIC | CONVERSION PROCEDURE |
|---|---|
| *r* (Bravais-Pearson Correlation) | no conversion needed |
| φ (Phi Coefficient) | no conversion needed |
| Cohen's *d* | esc::pearsons_r() |
| Odds Ratio | esc::pearsons_r() |
| η² (Eta squared) | esc::pearsons_r() |
| Cohen's *f* | esc::pearsons_r() |
| *R*² (R squared) | sqrt() |
| χ² (Chi squared) | no conversion |
| *b* (Standardized Regression Coefficient) | no conversion |
| Cramér's V | no conversion |
| Bayes Factor | no conversion |
| Hazard Ratio | no conversion |
| Cohen's *q* | no conversion |
| Risk Ratio | no conversion |
| $r_s$ (Spearman's Rho) | no conversion |
| rτ (Kendall's Tau) | no conversion |

**Table 1** Conversion of standardized effect sizes.

Effect sizes were coded as reported in the research articles (*reported effect sizes*) and remained unchanged. For *converted effect sizes*, original effect sizes were coded to be positive. To maintain uniformity of interpretation, replication effect sizes were matched so that positive values indicate effects in the same direction, while negative values indicate reversals (i.e., the replication study shows an effect size opposite to that of the original study). For example, if the original effect size was $r_{original} = .24$ and the replication effect was $r_{replication} = -.04$, no changes were made. If, however, $r_{original} = -.60$ and $r_{replication} = .01$, converted effect sizes were coded as $r_{original} = .60$ and $r_{replication} = -.01$.

## Submission of Individual Entries

Researchers can enter replication findings using two paths:

**(1)** An online submission form (https://www.soscisurvey.de/replicate) via SoSciSurvey (Leiner, 2019), which includes a tutorial video (https://osf.io/62cxy) in which researchers are exhaustively guided how to enter data (e.g., original and replication effect sizes, sample sizes, and descriptions of the entered findings; see Table 2 for all variables and which ones are mandatory for new entries). For the steps after the submission, we created an R code (https://osf.io/2rv9z) that downloads submitted entries, converts effect sizes, and transforms them into a format compatible with the database.

**(2)** A Google Sheets spreadsheet allows input of data in a publicly available document (https://docs.google.com/spreadsheets/d/1x68oW2H_Xrdv44fIeycl4fegsmQgCa60GxeZZ_hAR90/edit?pli=1#gid=1463805480). Variables are listed with brief descriptions, and mandatory variables are highlighted. After submission, contributors are prompted to contact the core team, who validate the new entry and copy it to the main dataset.

## Coded Variables

An overview of all variables included in the database is provided in Table 2.

| VARIABLE | DESCRIPTION | EXAMPLE VALUES AND NOTES | MANDATORY? |
|---|---|---|---|
| **id** | ID variable that is different for independent samples and constant for identical/dependent/overlapping samples. | "Soscisubmission47" for the 47th submission via the submission portal. | yes |
| **validated** | Has this entry been validated? NA = no 0 = no 1 = yes and everything is correct (corresponds to what is reported in the source) 2 = yes and errors were highlighted, corrected, and commented in notes_validation 3 = yes, errors have not yet been corrected 4 = [for individual submissions only] necessary values are present/data is sufficient for effect sizes calculation 5 = [for individual submissions only] some values are missing 6 = [for individual submissions only] data is complete and has been validated with respect to its sources (e.g., papers, datasets). | "1" In the long term, this should be "1" for all entries. | yes |
| **validated_person** | Who has checked the entry? (initials of the person's name) | "LK" | yes |
| **source** | Source of the entry; new additions are mostly coded as "Individual submissions". | "OSC 2015" for findings from the Open Science Collaboration, 2015. | yes |
| **discipline** | Which scientific discipline does the finding come from or in which has it been published? | "Applied Linguistics" | no |
| **effect** | What is the phenomenon or "effect" called? (e.g., "heat priming") Leave empty if there is no association with a family of phenomena. | "Chameleon effect" | no |
| **tags** | Tags to increase findability of the entry. | "Mimicry" | no |
| **description** | Description of the effect/phenomenon under investigation. | "People unconsciously imitate non-verbal behavior" | yes |
| **notes** | Notes for data entry. | Notes about imprecise reports, justifications for missing data, mentions of additional data that is not a replication but might be of interest for researchers investigating this phenomenon. | no |

(Contd.)

| VARIABLE | DESCRIPTION | EXAMPLE VALUES AND NOTES | MANDATORY? |
|---|---|---|---|
| contributors | For individual submissions: name of the person who submitted the effect to ReD. For all others: names of the contributors of the study. | "Leonard Kaiser" | only for individual submissions to allow checking |
| date_entered | Date of entry (dd.mm.yyyy); earliest entry is dated 01.01.2023. | "19.10.2023" | yes |
| notes_validation | Notes regarding the test. | "There are more effects in the original and the replication study which are not relevant to the main hypotheses." | no |
| exclusion | Reason for study exclusion. | "Same entry twice" | no |
| es_original | Original effect size converted to *r*. | Contributors were asked not to convert effect sizes themselves but to enter the unstandardized values or test statistics into the other variables. | |
| es_replication | Replication effect size converted to *r*. | See es_original | |
| n_original | Original study's sample size. | "100" | |
| n_replication | Replication study's sample size. | "150" | |
| ref_original | Reference (APA7 formatting) of the original study + Study number. | "Miller, D. T., & Ratner, R. K. (1998). The disparity between the actual and assumed power of self-interest. Journal of Personality and Social Psychology, 74(1), 53–62. https://doi.org/10.1037/0022-3514.74.1.53" | yes |
| doi_original | DOI for the reference of the original study (without "http" or "dx.doi.org"). | "https://doi.org/10.1037/0022-3514.74.1.53" | yes |
| ref_replication | Reference (APA7 formatting) of the replication study + Study number. | "Brick, C., Fillon, A., Yeung, S., Wang, M., Lyu, H., Ho, J., Wong, S. & Feldman, G. (2021). Self-interest is overestimated: Two successful pre-registered replications of Miller and Ratner (1998). Collabra Psychology. https://doi.org/10.1525/collabra.23443 Study 1" | yes |
| doi_replication | DOI for the reference of the replication study (without "http" or "dx.doi.org"). | "10.1525/collabra.23443" | yes |
| es_orig | Original effect size as formatted in the source materials (included for batch submissions and validation purposes; left empty for new submissions). | "d = 0.21" | no |
| es_rep | Replication effect size as formatted in the source materials (included for batch submissions and validation purposes; left empty for new submissions). | "d = 0.28" | no |
| es_orig_value | Original effect size value. | "3.13" | no |
| es_orig_estype | Original effect size type. | "OR" | no |
| es_rep_value | Replication effect size value. | "1.38" | no |
| es_rep_estype | Replication effect size type. | "OR" | no |
| es_orig_RRR | Obsolete variable, included for historic reasons. | | no |
| es_orig_RRR_estype | Obsolete variable, included for historic reasons. | | no |
| es_rep_RRR | Obsolete variable, included for historic reasons. | | no |
| es_rep_RRR_estype | Obsolete variable, included for historic reasons. | | no |
| osf_link | Link to the OSF project or to a repository that includes materials, data, and other relevant resources. | "https://osf.io/0aifq" | |
| outcome | Outcome of the replication study as coded in the subset of findings from curatescience.org (see also LeBel et al., 2018). | "No signal – inconsistent" | no |
| published_rep | Has the replication study been published? 0 = no 1 = as pre-print 2 = as peer-reviewed journal article 3 = as other (thesis, data set, …) | "2" | no |

(Contd.)

| VARIABLE | DESCRIPTION | EXAMPLE VALUES AND NOTES | MANDATORY? |
|---|---|---|---|
| **id_sample** | Unique ID per sample (if two effects originate from one sample, then enter the same values in each case). | "7a" and "7b" for two results from the same study but different sub-samples | yes |
| **same_design** | Was the same design used in the replication study? (e.g., within-subjects design, number of factors and factor levels, nesting, …) 0 = no, 1 = yes | "1" | no |
| **nesting** | Were all observations independent, nested, matched, clustered, …? | "Independent" | no |
| **same_test** | Was the same statistical test used in the replication study? (e.g., *t* test, ANOVA, …) 0 no, 1 = yes | "1" | no |
| **original_authors** | Were any of the original study's authors involved in the replication study? 0 = no, 1 = yes | "0" | no |
| **study_orig** | Number/sample/page of the original study. | Information about where to find the entered values. This should facilitate checking the entries | no |
| **study_rep** | Number/sample/page of the replication study. | Information about where to find the entered values. This should facilitate checking the entries | no |
| **teststatistic_orig** | Complete test statistic for the original finding. | "F(1,105) = 2.45, p = 0.12, etasq = 0.02" | no |
| **teststatistic_rep** | Complete test statistic for the replication finding. | "F(1,81) = 2.164, p = 0.145, etasq = 0.026" | no |
| **p_es_orig** | Page number on which the original effect size can be found in the publication of the original study. | "Page 2 (original study) Page 7 (replication study)" | no |
| **p_es_rep** | Page number on which the original effect size can be found in the publication of the replication study. | | no |
| **p_n_orig** | Page number on which the original sample size can be found in the publication of the original study. | | no |
| **p_n_rep** | Page number on which the original sample size can be found in the publication of the replication study. | | no |
| **result** | Result of the respective replication test. Success: Original and replication effect were both significant or both non-significant and effect sizes were in the same direction (if applicable). Informative failure to replicate: The condition for success is not met. This can be due to the effect being in the same direction but not significant (e.g., due to a lack of precision in the measurements), a significant effect in the opposite direction, or a null effect. Practical failure to replicate: Reporting beyond significance testing indicated that reasons other than effect sizes led to the replication study not being interpretable (e.g., the target sample size was not reached, the study had to be discontinued). Inconclusive: Reporting beyond significance testing indicated that the result is unclear (e.g., there were multiple tests, and some were successful and some were not, the hypothesis is not sufficiently specific). Mixed [only on aggregated levels and auto-coded]: When all replication findings for one original result are considered, results were not the same for all attempts. | "Success" | yes |
| **preregistration** | Link to the preregistration. | "https://osf.io/avf49" | no |
| **closeness_instructions** | Closeness between the original study and replication study regarding instructions. See also Replication Recipe; 1 = exact, 2 = close, 3 = different, 4 = does not apply, 5 = unknown. | 1 | no |
| **closeness_measures** | Closeness between the original study and replication study regarding measures. See also Replication Recipe; 1 = exact, 2 = close, 3 = different, 4 = does not apply, 5 = unknown. | 3 | no |

(Contd.)

Röseler et al. *Journal of Open Psychology Data* DOI: 10.5334/jopd.101

| VARIABLE | DESCRIPTION | EXAMPLE VALUES AND NOTES | MANDATORY? |
|---|---|---|---|
| **closeness_stimuli** | Closeness between the original study and replication study regarding stimuli. See also Replication Recipe; 1 = exact, 2 = close, 3 = different, 4 = does not apply, 5 = unknown. | 3 | no |
| **closeness_procedure** | Closeness between the original study and replication study regarding the procedure. See also Replication Recipe; 1 = exact, 2 = close, 3 = different, 4 = does not apply, 5 = unknown. | 2 | no |
| **closeness_location** | Closeness between the original study and replication study regarding the location where the study was conducted (e.g., city-country-continent, lab vs. field). See also Replication Recipe; 1 = exact, 2 = close, 3 = different, 4 = does not apply, 5 = unknown. | 1 | no |
| **closeness_renumeration** | Closeness between the original study and replication study regarding remuneration (e.g., payment, feedback on personal data such as IQ values, course credit). See also Replication Recipe; 1 = exact, 2 = close, 3 = different, 4 = does not apply, 5 = unknown. | 2 | no |
| **closeness_participants** | Closeness between the original study and replication study regarding participants (e.g., convenience sample, student sample, clickworkers). See also Replication Recipe; 1 = exact, 2 = close, 3 = different, 4 = does not apply, 5 = unknown. | 2 | no |
| **closeness_exclusions** | Closeness between the original study and replication study regarding exclusions. See also Replication Recipe; 1 = exact, 2 = close, 3 = different, 4 = does not apply, 5 = unknown. | 3 | no |
| **closeness_language** | Closeness between the original study and replication study regarding language. See also Replication Recipe; 1 = exact, 2 = close, 3 = different, 4 = does not apply, 5 = unknown. | 4 | no |
| **closeness_nationality** | Closeness between the original study and replication study regarding nationality. See also Replication Recipe; 1 = exact, 2 = close, 3 = different, 4 = does not apply, 5 = unknown. | 2 | no |
| **differences** | Specification of all differences between the original study and the replication written in bullet points or plain text. | "The original study had an additional condition that was not included in the replication study. Also, the original study was in Dutch and English, the replication was in German." | no |
| **vi_orig** | Variances of original effects, automatically computed. | "0.0181092" | no |
| **vi_rep** | Variances of replication effects, automatically computed. | "0.00945695" | no |
| **ci.lower_original** | Lower confidence interval for the standardized effect size (replication effect), automatically computed. | "–0.0497735" | no |
| **ci.upper_original** | Upper confidence interval for the standardized effect size (replication effect), automatically computed. | "0.32261748" | no |
| **ci.lower_replication** | Lower confidence interval for the standardized effect size (replication effect), automatically computed. | "–0.0564059" | no |
| **ci.upper_replication** | Upper confidence interval for the standardized effect size (replication effect), automatically computed. | "0.36426571" | no |
| **significant_original** | Was the original effect significant ($\alpha = .05$)? 1 = yes, 0 = no, automatically computed. | "0" | no |
| **significant_replication** | Was the replication effect significant ($\alpha = .05$)? 1 = yes, 0 = no, automatically computed. | "0" | no |
| **power** | Replication study power based on replication $N$ and original effect size converted to $r$, automatically computed. | "0.358" | no |
| **orig_journal** | Journal that published the original findings. | "Scientific Reports" | no |

**Table 2** Overview of variables included in the dataset.

## 2.2 TIME OF DATA COLLECTION

The database as of October 2023 contains results from original studies that have been published between 1935 (Stroop, 1935) and 2023 (e.g., Röseler, Doetsch, et al., 2023). Like in most meta-analytical datasets, data collection times for the included studies are mostly unknown and only publication years are provided.

Collection of meta-data is ongoing and will continue for the foreseeable future (e.g., via hackathons and workshops at conferences, collaborations with large-scale projects, and literature alerts). After collecting the currently hosted data, aggregating and formatting of the datasets began in May 2022 using the Open Science Framework Registries webpage (https://www.osf.io/registries).

## 2.3 LOCATION OF DATA COLLECTION

Worldwide/asynchronously/remote.

## 2.4 SAMPLING, SAMPLE AND DATA COLLECTION

The presented dataset represents the Replication Database dated 16th October 2023 and consists of multiple sub-datasets and individual replications. Historically, the basis was formed by an aggregation of data from OSF's registries (Röseler et al., 2022) and replications conducted by Feldman and colleagues ("Collaborative Open-science and meta REsearch, CORE", CORE Team, 2024). We then added large-scale projects, such as data from the Reproducibility Project Psychology (Open Science Collaboration, 2015) and others. All further entries that we had to code manually were labeled as individual submissions. These include data from CurateScience.org or specific journal issues dedicated to replications. We issued a call for results (https://osf.io/v4xjk) via 14 channels (i.e., conferences, social networks, and mailing lists) in March and April 2023 (for an overview see https://osf.io/d5r7c). Since then, project leads and research assistants have been manually coding studies from further lists, databases, and literature searches. We have also been reaching out to large-scale replication projects and asked them to help add their data. In late 2023, the Replication Database and the Framework for Open and Reproducible Research Training (FORRT) Replications and Reversals project joined forces, with a merging of the two databases taking place until late 2024. In parallel, we have been validating entries submitted by other researchers. An overview of data sources and distributions of the original publications throughout the years is provided in Tables 3–5 and Figures 3–4. Dataset descriptions and plots were created with R version 4.3.2 (R Core Team, 2018) and the *packages* ggplot2 (Wickham, 2016), *openxlsx* (Schauberger & Walker, 2021), and *psych* (Revelle, 2024). Code to reproduce the results is available online (https://osf.io/j8qav).

In total, there are 1,239 entries (i.e., pairs of original and replication effects). Note that effect sizes and sample sizes could not be coded for 201 cases. The entries stem from 336 independent original studies and 468 independent replication findings. With independent, we refer to non-overlapping samples. For example, research articles reported results from up to 80 independent studies (see also Table 3 for a summary).

Replication outcomes were taken from the reported replications in the OSF registries, coded from author statements, and computed from reported effect sizes in some cases. Most findings were informative failures to replicate ($k = 641$) followed by successes ($k = 447$). Assessments could not be made for $k = 133$ findings, $k = 15$ were inconclusive, and $k = 3$ entries were practical failures to replicate (see also Table 4 for definitions of outcomes).

| CATEGORY | VALUE |
|---|---|
| All Entries | 1,239 |
| Independent Original Studies | 336 |
| Independent Replication Findings | 468 |
| Entries Not Included in Quantitative Analyses | 201 |

**Table 3** Description of entries from the Replication Database.

| OUTCOME | NUMBER OF ENTRIES | DEFINITION OF OUTCOME |
|---|---|---|
| Inconclusive | 15 | Reporting beyond significance testing indicated that the result is unclear (e.g., there were multiple tests, and some were successful and some were not, the hypothesis is not sufficiently specific). |
| Informative Failure to Replicate | 641 | The condition for success is not met. This can be due to the effect being in the same direction but not significant (e.g., due to a lack of precision in the measurements), a significant effect in the opposite direction, or a null effect. |
| Practical Failure to Replicate | 3 | Reporting beyond significance testing indicated that reasons other than effect sizes led to the replication study not being interpretable (e.g., the target sample size was not reached, the study had to be discontinued). |
| Success | 447 | Original and replication effect were both significant or both non-significant and effect sizes were in the same direction (if applicable). |
| Not Available | 133 | No assessment of outcome has been coded (e.g., due to missing original or replication effect size or sample sizes or no clear evaluation in the replication report). |

**Table 4** Replication outcomes.

Data from the original projects (e.g., Open Science Collaboration, 2015) have been reformatted. In some cases, effect sizes have been standardized, and most references have been added (original materials mostly included short references without DOIs, only author names, or references in formats other than APA). Further, we added variables such as journals that published the original findings, 95% confidence intervals for original and replication effect sizes, outcomes, and replication study power. An overview of the number of effect sizes by source is provided in Table 5.

| SOURCE | NUMBER OF EFFECT SIZES |
|---|---|
| CORE | 109 |
| CRSP Special Issue | 4 |
| Individual Submissions | 247 |
| ML1 | 352 |
| ML3 | 145 |
| OSC 2015 | 167 |
| OSF Registries | 95 |
| RRR | 120 |

**Table 5** Sources of replication findings.

*Note.* CORE = Collaborative Open-science and meta REsearch (CORE Team, 2024), CRSP = Comprehensive Results in Social Psychology (Journal), ML = Many Labs (e.g., Klein et al., 2014), OSC = Open Science Collaboration (2015), OSF = Open Science Framework (https://osf.io), RRR = Registered Replication Report (e.g., Hagger et al., 2016; O'Donnell et al., 2018).

On average, replication effect sizes were smaller than original effect sizes. Replication effect sizes divided by original effect sizes ($k$ = 1,050, M = 0.52, SD = 0.98, Min = –6.9, Max = 22.82, Md = 0.387; excluding cases with original effect sizes of 0). Figure 3 provides a scatterplot of original and replication effect sizes in the style of Open Science Collaboration (2015). An interactive version with an up-to-date dataset is available online (https://forrt-replications.shinyapps.io/fred_explorer). The distribution of relative effect sizes is displayed in Figure 4.

## 2.5 MATERIALS/SURVEY INSTRUMENTS

- Call for Results: https://osf.io/v4xjk
- Instructions for coding: https://osf.io/47zwe
- Instructions for validating: https://osf.io/y3fm8
- Submission form: https://osf.io/q5hfj (archived) or https://www.soscisurvey.de/replicate (non-permanent link)

## 2.6 QUALITY CONTROL
### Validation for Individual Submissions

As a collaborative community effort from the contributors, all mandatory fields (Table 2) were systematically verified by one person per entry (listed in the variable validated_person). These seven contributors were students fulfilling course credits or research assistants. They were acquainted with statistical methods (e.g., effect sizes and null hypothesis significance testing) and used standardized instructions (https://osf.io/y3fm8). For example, they

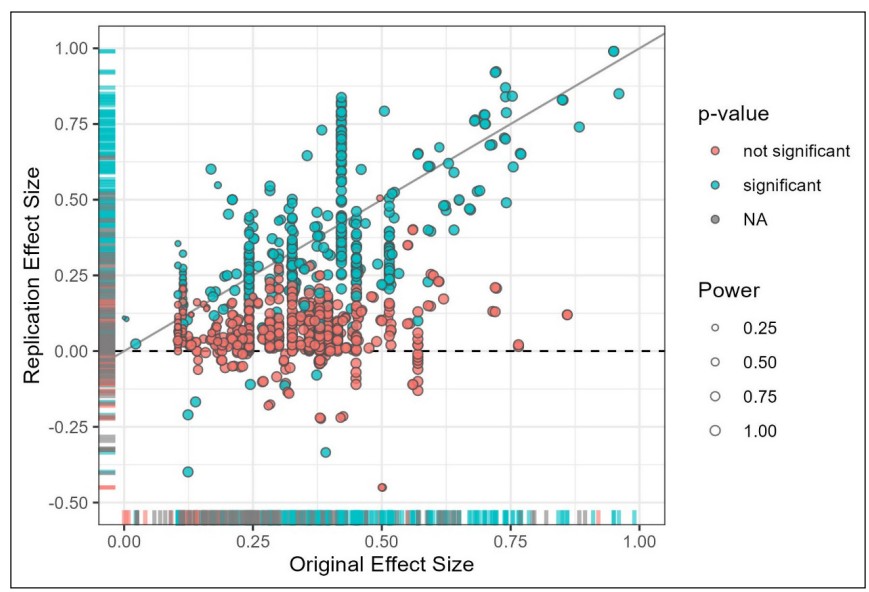

**Figure 3** Original and replication effect size by significance of replication effect and power of the replication study.

*Note.* $k$ = 1,051 pairs of original and replication effect sizes converted to correlation coefficients. Some code for the plot was taken from Open Science Collaboration (2015). Power: Statistical power of the replication study given the replication sample size and the original effect size. P-value of the replication study was estimated based on converted effect sizes and may be skewed for nested designs (α = 5%). Points on the diagonal solid line reflect cases where replication effect size = original effect size. Points on the horizontal dashed line represent replication effect sizes close to 0.

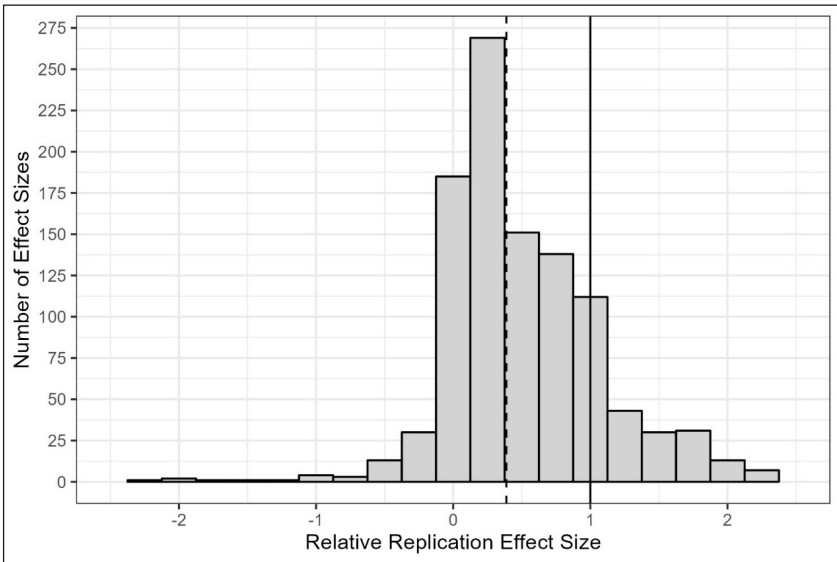

**Figure 4** Histogram of relative replication effect sizes.

*Note*. X-axis was truncated for readability and some relative replication effect sizes are not visible. The dashed line represents the median of 0.387, $k$ = 1,050. The solid line represents 1, that is, the relative effect size that results from both effects being the same. Cases where the original effect size was zero were removed due to the ratio yielding infinity.

tested hyperlinks, and assessed descriptions and keywords for plausibility. The attribution of effects to one or multiple samples and the accurate naming of Sample IDs were also examined.

The person indicated by the variable validated_person scrutinized both the original and replication papers to ensure the congruence of reported sample sizes with the submitted information. Special focus was placed on the accuracy of sample sizes with regard to the removal of participants. Additionally, the effect sizes and their types were individually examined in both the original and replication papers. In case of uncertainties encountered during these steps, we contacted the contributor of the results for further clarification and LR was informed of the potential problems.

### Validation for Batch Submissions

With batch submissions, we refer to submissions of many findings at once, such is the case for large-scale projects (e.g., Open Science Collaboration, 2015). In these cases, the original dataset was converted and entered in the database. For each batch of submissions, a project team member checked whether the entries regarding effect sizes, sample sizes, and references in the Replication Database aligned with those of the submitted studies. This work was again done by research assistants or the project lead. In some cases, the original authors of large datasets validated the entries or converted the data.

### Dealing with Inconsistencies

In cases of inconsistencies, we corrected values to match the source material. We identified an error in a replication report, confirmed it with the author(s), and commented

on PubPeer. If authors were unreachable, we relied on the original or replication reports. After other researchers flagged two errors in the CurateScience data (LeBel et al., 2018), we revalidated all CurateScience entries by comparing effect and sample sizes directly with the original reports rather than the database. For future errors in our database, we encourage researchers to submit a comment to this article via PubPeer (https://pubpeer.com).

### Limitations

Several limitations arise due to the large size of the database, limited resources, collaborative data collection, and ongoing discussions about replication methodology.

- Deprecation of entries: Variables such as publication status may change over time from "pre-print" to "journal article". Although we ask all contributors to let us know if variables change, there is currently no procedure to guarantee that this variable is up to date.
- Outcome variables: There are numerous ways to measure replication outcomes with regard to the original study's findings. Effect sizes or relative effect sizes are the most fine-grained way to code outcomes while also being able to compare them but some researchers or practitioners may prefer categorical values such as success or failure. Although the database includes the evaluations suggested by Brandt et al. (2014), the current coding scheme is inconsistent as some entries were taken from what replicators coded in the OSF registries when publishing result reports using the Replication Recipe post-completion template (Brandt et al., 2014) and some were computed based on the entered effect sizes or were filled out by contributors of the findings who

would otherwise not have categorized the replication attempt using these labels. Note that more objective classifications such as suggested by LeBel et al. (2019) can be computed based on the present values (e.g., signal vs. no signal, direction).

- Replication closeness: As described above, replication closeness is difficult to measure, hard to validate, and should be used with caution. Currently, coding replication closeness is optional, which is why it is also missing for a large proportion of entries.
- Ignorance of nested designs: Although commensurability of different effect sizes is statistically possible through conversion, caution should be exercised when interpreting effects from between-subject designs compared to those from within-subject or nested designs as estimates such as significance level or power will be skewed. Note, however, that the design has been coded and cases can be filtered for it.
- Quality control: Due to crowdsourcing and limited resources, the dataset is likely to contain errors. In the trade-off between comprehensiveness and correctness, we strive for the former to maximize visibility and findability of replications. For better or worse, researchers can easily go from our database to the original reports. Data from large scale projects was only compared to their data as not every single study could be checked. Checks do not include reproductions of analyses but only comparison of values. In many cases, we noticed discrepancies between entered sample sizes and degrees of freedom from the respective tests as researchers entered the total sample size and not the sample size used for the respective tests. For individual submissions, we reached out to the contributors and could resolve all inconsistencies.
- Coding of samples, studies, dependent variables, and items: Entries are coded so that dependent samples (i.e., samples that belong to the same replication study but were used to replicate different original findings) and study numbers from original and replication findings can be identified. However, there is no standardized procedure to code hypotheses, dependent variables, or items. These are usually collapsed in the description but future research or a revision of the database may benefit from a more differentiated coding procedure.

## 2.7 DATA ANONYMIZATION AND ETHICAL ISSUES
Because all entries concern scientific contributions such as research articles or datasets, we did not anonymize the data.

## 2.8 EXISTING USE OF DATA
Subsets of the data (e.g., data from Many Labs) or aggregated versions have been used for meta-research

(e.g., Sotola, 2023). At the time of publication, we are aware of two projects that have used the entire database.

- Röseler, L. (2023, October 16). *Predicting Replication Rates with Z-curve: A Brief Exploratory Validation Study Using the Replication Database.* https://doi.org/10.31222/osf.io/ewb2t
- Röseler, L., Doetsch, C. A., Kaiser, L. D., Gendlina, T. D., Klett, N., Krapp, J., Seida, C., Förster, N., & Schütz, A. (2023, March). *The Replication Database: Making transparent what replicated and what did not.* Presentation at the Conference for Experimental Psychology (TeaP), Trier, Germany. https://osf.io/sf8j2

# (3) DATASET DESCRIPTION AND ACCESS

The datasets and materials are openly available in the OSF repository (https://osf.io/9r62x/) and will be updated continuously as the database grows.

- Dataset used for the reported analyses: https://osf.io/qtkzy
- Google Sheets spreadsheet that we plan to update regularly: https://docs.google.com/spreadsheets/d/1x68oW2H_Xrdv44fIeycl4fegsmQgCa60GxeZZ_hAR90/edit?pli=1#gid=1463805480 (non-permanent link)
- Interactive Shiny Application: https://forrt-replications.shinyapps.io/fred_explorer (non-permanent link)
- Interactive Shiny Application for Reference List Annotation: https://forrt-replications.shinyapps.io/fred_annotator (non-permanent link)
- Dataset changelog (starting January 2024): https://osf.io/ej46t

### 3.1 REPOSITORY LOCATION
Repository link: https://osf.io/9r62x
Frozen Repository as of August 2024: https://osf.io/c9rny
Repository DOI: https://doi.org/10.17605/OSF.IO/9R62X

### 3.2 OBJECT/FILE NAME
Reported version: https://osf.io/qtkzy
Most recent version: "FReD.xlsx" available at https://osf.io/z5u9b

### 3.3 DATA TYPE
Secondary data, processed data, aggregated data.

### 3.4 FORMAT NAMES AND VERSIONS
Datasets are available in .csv and .xlsx formats.

### 3.5 LANGUAGE
English, German (variable labels).

### 3.6 LICENSE
CC-By Attribution 4.0 International.

### 3.7 LIMITS TO SHARING

The data is not under embargo. It contains the names of researchers who conducted original studies and replication studies (i.e., references) and the names of researchers who contributed to the dataset. The data may be updated with further replication findings and we plan to maintain and extend the Shiny Application for several more years.

Please cite this article and along with it the most recent version of the OSF-project (https://osf.io/9r62x) that includes a version number and contributors who joined the project since 04/2023.

### 3.8 PUBLICATION DATE

An initial version of the dataset has been shared on 22/01/2023, on the Open Science Framework (OSF; https://osf.io/2a3gb). The reported results are based on the version from 16/10/2023.

### 3.9 FAIR DATA/CODEBOOK

We have posted the dataset publicly on the OSF (https://osf.io/9r62x). We provide coding instructions as text (https://osf.io/hvebr) and as a video (https://osf.io/tvh9n). The OSF project has been assigned a DOI (https://doi.org/10.17605/OSF.IO/9R62X). Code that formats data from the submission portal to match the structure of the dataset is available online (https://osf.io/uzpgb) and can be run with open-source software (e.g., GNU-R, R Core Team, 2018).

# (4) REUSE POTENTIAL

We encourage others to use the Replication Database for their research or for educational purposes, add their replication findings to the database, or merge it with other existing databases. We suggest using it for a wide variety of different purposes.

- **Increase findability of replications**: Researchers, teachers, policy-makers, and professionals often rely on scientific evidence. With the database, they can easily and quickly get an overview of the potential robustness, generalizability, and heterogeneity in effects.
- **Summarize replication efforts by area**: The dataset can be used to summarize the robustness of findings by disciplines, research areas, phenomena, journals, time of publication, or researchers. This way, researchers can identify areas where replications are common or uncommon, which may aid in planning replication attempts, monitoring replication affinity, or determining directions of future research. For example, if for a phenomenon, some replications are successful and others are not, they can be compared and reveal potentially relevant background variables.

- **Inclusion in traditional meta-analyses**: With meta-analyses often struggling to include unpublished findings, replications, and null-findings, we believe that the Replication Database as a low-threshold opportunity to publish replication attempts can help researchers find studies that they can include in their meta-analyses and that may correct for the publication bias.
- **Validation data for bias-correction methods**: Methods that predict replication rates or correct meta-analytical effect sizes for publication bias and questionable research practices are often evaluated using simulated data (e.g., Carter et al., 2019) and validations with existing data need to rely on few and scattered large-scale projects (e.g., Sotola & Credé, 2022). With the replication database, these proposed methods can easily be tested against a large set of real data. In turn, the dataset can inform simulation studies about characteristics of replication studies from different research areas in psychology.
- **Inform replication guidelines**: With replication guidelines still being developed, we believe that the Replication Database can support the development of evidence-based replication guidelines and evaluation protocols. For example, if certain features of replication studies affect replication outcomes positively (e.g., preregistration of the study's methods and analysis plan), recommendations to preregister replication studies can rest on this evidence.
- **Teaching**: At the moment, textbooks and teaching materials are highly likely to include findings that could not be replicated. In the past, problems regarding findability of replication attempts made it difficult to provide a more nuanced discussion. The Replication Database can help researchers revising these materials and including more recent findings for the discussed phenomena or theories via a reference list annotation tool. This way, references can be read and annotated with respect to replication attempts (e.g., if there have been any replication attempts and what their outcomes were).

Moreover, instead of relying on singular findings, teachers and lecturers can for example ask students to examine replications, compare them with the original findings, and thereby help them develop skills to critically evaluate bodies of research.

Finally, replication studies have become an integral part of undergraduate research (Boyce et al., 2023; Jekel et al., 2020; Korell et al., 2023; Quintana, 2021). The database provides a low-threshold opportunity to make student replications visible.

We invite researchers to join our effort to make replications in psychological science and beyond transparent in a systematic manner.

## ACKNOWLEDGEMENTS

We thank the Open Science Collaboration, Etienne LeBel (0000-0001-7377-008X), Aaron Charlton (0000-0001-8384-3852), all members of the FORRT (0000-0002-7562-5342) Replications and Reversals community, the Replication Database community, and student researchers that contributed to the first coded set of replication findings, that is, Taisia Gendlina, Josefine Krapp, Noemi Labusch, and Anja Wagner for the effort that went into their respective lists of replication studies that helped us to accumulate replication findings. We also thank numerous anonymous researchers for coding or submitting replication results. Finally, we thank the Society for the Improvement for Psychological Science (SIPS), the Psychological Science Accelerator (PSA), the Advancing Big-team Reproducible Science through Increased Representation (ABRIR) project, and the Association for Interdisciplinary Meta-Research and Open Science (AIMOS) for allowing us to host multiple hackathons to shape this database.

## FUNDING INFORMATION

We acknowledge support by the Open Access Publication Fund of University of Münster. Parts of this project were furthermore supported through a grant from the University of Bamberg's Interne Forschungsförderung allocated to Lukas Röseler, by a grant from the Nederlandse Organisatie voor Wetenschappelijk's (NWO) Open Science Fund allocated to Flavio Azevedo, Helena Hartmann, Leticia Micheli, Lukas Wallrich, and Sam Parsons (see https://www.researchequals.com/api/modules/main/r4qf-7peg), and by a grant from the German Research Foundation (DFG; project number: 497678237) allocated to Oliver Genschow and the Leuphana University Lüneburg. Helena Hartmann was supported by the Deutsche Forschungsgemeinschaft (DFG, German Research Foundation – Project-ID 422744262 – TRR 289). Zoran Pavlović was supported by the Ministry of Science, Technological Development, and Innovation of the Republic of Serbia (contract number 451–03-66/2024-03/200163). Robert Ross was supported by the Australian Research Council (grant number: DP180102384) and the John Templeton Foundation (grant ID: 62631). Manuel Rausch was supported by Grant RA2988/4-1 by the Deutsche Forschungsgemeinschaft. Maria Montefinese was supported by the Investment line 1.2 "Funding projects presented by young researchers" (CHILDCONTROL) from the European Union – NextGenerationEU. J. Lukas Thürmer's contribution was funded in part by the Austrian Science Fund (FWF) [https://dx.doi.org/10.55776/P37261] and by an EASP pre-registered grant. Matthias F. J. Sperl was supported by the "Justus Liebig University Postdoc Fund," provided by the Postdoc Career and Mentoring Office of the University of Giessen (Germany).

The funders had no role in data collection and analysis, decision to publish, or preparation of the manuscript.

## COMPETING INTERESTS

The author(s) declare no conflict of interest associated with the publication of this manuscript. TE is Co-Editor-in-Chief for the JOPD and has not had any editorial input into the decisions made on the manuscript.

## AUTHOR CONTRIBUTIONS

**Lukas Röseler** – Conceptualization, Data Curation, Formal Analysis, Investigation, Funding Acquisition, Methodology, Project Administration, Resources, Software, Supervision, Validation, Visualization, Writing – Original Draft, Writing – Review and Editing

**Leonard Kaiser** – Data Curation, Validation, Investigation, Writing – Original Draft, Writing – Review and Editing

**Christopher Doetsch** – Data Curation, Validation, Investigation, Writing – Original Draft, Writing – Review and Editing

**Noah Klett** – Data Curation, Formal Analysis, Investigation, Software, Writing – Review and Editing

**Christian Seida** – Data Curation, Formal Analysis, Investigation, Software, Writing – Review and Editing

**Astrid Schütz** – Resources, Funding Acquisition, Writing – Review and Editing

**Balazs Aczel** – Resources, Writing – Review and Editing

**Nadia Adelina** – Resources, Writing – Review and Editing

**Valeria Agostini** – Resources, Writing – Review and Editing

**Samuel Alarie** – Resources, Writing – Review and Editing

**Nihan Albayrak-Aydemir** – Resources, Writing – Review and Editing

**Alaa Aldoh** – Resources, Writing – Review and Editing

**Ali H. Al-Hoorie** – Resources, Writing – Review and Editing

**Flavio Azevedo** – Resources, Writing – Review and Editing

**Bradley J. Baker** – Resources, Writing – Review and Editing

**Charlotte Lilian Barth** – Data Curation, Writing – Review and Editing

**Julia Beitner** – Resources, Writing – Review and Editing

**Cameron Brick** – Resources, Writing – Review and Editing

**Hilmar Brohmer** – Resources, Writing – Review and Editing

**Subramanya Prasad Chandrashekar** – Resources, Writing – Review and Editing

**Kai Li Chung** – Resources, Writing – Review and Editing

**Jamie P. Cockcroft** – Resources, Writing – Review and Editing

**Jamie Cummins** – Resources, Writing – Review and Editing

**Veronica Diveica** – Resources, Writing – Review and Editing

**Tsvetomira Dumbalska** – Resources, Writing – Review and Editing

**Emir Efendic** – Resources, Writing – Review and Editing

**Mahmoud Elsherif** – Resources, Writing – Review and Editing

**Thomas Evans** – Resources, Writing – Review and Editing

**Gilad Feldman** – Resources, Writing – Review and Editing

**Adrien Fillon** – Resources, Writing – Review and Editing

**Nico Förster** – Resources, Writing – Review and Editing

**Joris Frese** – Resources, Writing – Review and Editing

**Oliver Genschow** – Resources, Writing – Review and Editing

**Vaitsa Giannouli** – Resources, Writing – Review and Editing

**Biljana Gjoneska** – Resources, Writing – Review

**Timo Gnambs** – Resources, Writing – Review and Editing

**Amélie Gourdon-Kanhukamwe** – Resources, Writing – Review and Editing

**Christopher J. Graham** – Resources, Writing – Review and Editing

**Helena Hartmann** – Resources, Writing – Review and Editing

**Clove Haviva** – Resources, Writing – Review and Editing

**Alina Herderich** – Resources, Writing – Review and Editing

**Leon P. Hilbert** – Resources, Writing – Review and Editing

**Darías Holgado** – Resources, Writing – Review and Editing

**Ian Hussey** – Resources, Writing – Review and Editing

**Zlatomira G. Ilchovska** – Resources, Writing – Review and Editing

**Tamara Kalandadze** – Resources, Writing – Review and Editing

**Veli-Matti Karhulahti** – Resources, Writing – Review and Editing

**Leon Kasseckert** – Data Curation, Validation

**Maren Klingelhöfer-Jens** – Resources, Writing – Review and Editing

**Alina Koppold** – Resources, Writing – Review and Editing

**Max Korbmacher** – Resources, Writing – Review and Editing

**Louisa Kulke** – Resources, Writing – Review and Editing

**Niclas Kuper** – Resources, Writing – Review and Editing

**Annalise LaPlume** – Resources, Writing – Review and Editing

**Gavin Leech** – Resources, Writing – Review and Editing

**Feline Lohkamp** – Data Curation, Writing – Review and Editing

**Nigel Mantou Lou** – Resources, Writing – Review and Editing

**Dermot Lynott** – Resources, Writing – Review and Editing

**Maximilian Maier** – Resources, Writing – Review and Editing

**Maria Meier** – Resources, Writing – Review and Editing

**Maria Montefinese** – Resources, Writing – Review and Editing

**David Moreau** – Resources, Writing – Review and Editing

**Kellen Mrkva** – Resources, Writing – Review and Editing

**Monika Nemcova** – Resources, Writing – Review and Editing

**Danna Oomen** – Supervision, Writing – Review and Editing

**Julian Packheiser** – Resources, Writing – Review and Editing

**Shubham Pandey** – Resources, Writing – Review and Editing

**Frank Papenmeier** – Resources, Writing – Review and Editing

**Mariola Paruzel-Czachura** – Resources, Writing – Review and Editing

**Yuri G. Pavlov** – Resources, Writing – Review and Editing

**Zoran Pavlović** – Resources, Writing – Review and Editing

**Charlotte R. Pennington** – Resources, Writing – Review and Editing

**Merle-Marie Pittelkow** – Resources, Writing – Review

**Willemijn Plomp** – Resources, Writing – Review and Editing

**Paul E. Plonski** – Resources, Writing – Review and Editing

**Ekaterina Pronizius** – Resources, Writing – Review and Editing

**Andrew Adrian Pua** – Resources, Writing – Review and Editing

**Katarzyna Pypno-Blajda** – Resources, Writing – Review and Editing

**Manuel Rausch** – Resources, Writing – Review and Editing

**Tobias R. Rebholz** – Resources, Writing – Review and Editing

**Elena Richert** – Resources, Writing – Review and Editing

**Jan Philipp Röer** – Resources, Writing – Review and Editing

**Robert Ross** – Resources, Writing – Review and Editing

**Kathleen Schmidt** – Resources, Writing – Review and Editing

**Aleksandrina Skvortsova** – Resources, Writing – Review and Editing

**Matthias F. J. Sperl** – Resources, Writing – Review and Editing

**Alvin W. M. Tan** – Resources, Writing – Review and Editing

**J. Lukas Thürmer** – Resources, Writing – Review and Editing

**Aleksandra Tołopiło** – Resources, Writing – Review and Editing

**Wolf Vanpaemel** – Resources, Writing – Review and Editing

**Leigh Ann Vaughn** – Resources, Writing – Review and Editing

**Steven Verheyen** – Resources, Writing – Review and Editing

**Lukas Wallrich** – Resources, Writing – Review and Editing

**Lucia Weber** – Data curation, Writing – Review and Editing

**Julia K. Wolska** – Resources, Writing – Review and Editing

**Mirela Zaneva** – Resources, Writing – Review and Editing

**Yikang Zhang** – Resources, Writing – Review and Editing

*Note.* Contributors of database entries received the CRediT role "Resources". Contributors coding variables or converting values were assigned the CRediT role "Data Curation".

Please note that to write a static report, the up-to-date database is necessarily larger and mistakes in the present version have been corrected for more recent versions. We took some of the text for this manuscript from our previous dataset publication at the JOPD (https://openpsychologydata.metajnl.com/articles/10.5334/jopd.67).

## AUTHOR AFFILIATIONS

**Lukas Röseler** orcid.org/0000-0002-6446-1901
Münster Center for Open Science, University of Münster, Germany; Institute of Psychology, University of Bamberg, Germany

**Leonard Kaiser**
University of Bamberg, Germany

**Christopher Doetsch**
University of Bamberg, Germany

**Noah Klett**
University of Bamberg, Germany

**Christian Seida** orcid.org/0000-0001-8884-2736
University of Bamberg, Germany

**Astrid Schütz** orcid.org/0000-0002-6358-167X
University of Bamberg, Germany

**Balazs Aczel** orcid.org/0000-0001-9364-4988
ELTE Eotvos Lorand University, Hungary

**Nadia Adelina** orcid.org/0000-0002-8808-2439
University of Hong Kong, Hong Kong SAR

**Valeria Agostini** orcid.org/0000-0003-0314-2998
University of Birmingham, United Kingdom

**Samuel Alarie**
University of Montreal, Canada

**Nihan Albayrak-Aydemir** orcid.org/0000-0003-3412-4311
Boğaziçi University, Türkiye; London School of Economics and Political Science, United Kingdom

**Alaa Aldoh** orcid.org/0000-0003-1988-0661
University of Amsterdam, the Netherlands

**Ali H. Al-Hoorie** orcid.org/0000-0003-3810-5978
Royal Commission for Jubail and Yanbu, Saudi Arabia

**Flavio Azevedo** orcid.org/0000-0001-9000-8513
Utrecht University, the Netherlands

**Bradley J. Baker** orcid.org/0000-0002-1697-4198
Temple University, United States

**Charlotte Lilian Barth**
Leuphana University, Germany

**Julia Beitner** orcid.org/0000-0002-2539-7011
Goethe University Frankfurt, Germany

**Cameron Brick** orcid.org/0000-0002-7174-8193
Department of Psychology, University of Amsterdam, the Netherlands

**Hilmar Brohmer** orcid.org/0000-0001-7763-4229
Department of Psychology, University of Graz, Austria

**Subramanya Prasad Chandrashekar** orcid.org/0000-0002-8599-9241
Department of Psychology, NTNU-Norwegian University of Science and Technology, Norway

**Kai Li Chung** orcid.org/0000-0003-0012-8752
University of Nottingham Malaysia, Malaysia; University of Reading Malaysia, Malaysia

**Jamie P. Cockcroft** orcid.org/0000-0002-0637-8851
Department of Psychology, University of York, United Kingdom

**Jamie Cummins** orcid.org/0000-0002-9729-4900
Institute of Marketing and Business Administration & Institute of Psychology, University of Bern, Switzerland

**Veronica Diveica** orcid.org/0000-0002-5696-8200
Department of Neurology and Neurosurgery, Montreal Neurological Institute, McGill University, Canada

**Tsvetomira Dumbalska** orcid.org/0000-0002-5761-8536
Department of Experimental Psychology, University of Oxford, United Kingdom

**Emir Efendic** orcid.org/0000-0002-2365-0247
Maastricht University, the Netherlands

**Mahmoud Elsherif** orcid.org/0000-0002-0540-3998
Department of Psychology, University of Birmingham, United Kingdom

**Thomas Evans** orcid.org/0000-0002-6670-0718
School of Human Sciences and Institute for Lifecourse Development, University of Greenwich, United Kingdom

**Gilad Feldman** orcid.org/0000-0003-2812-6599
University of Hong Kong, Hong Kong SAR

**Adrien Fillon** orcid.org/0000-0001-8324-2715
SInnoPSis, University of Cyprus, Cyprus

**Nico Förster** orcid.org/0009-0005-6312-3096
RPTU Kaiserslautern-Landau, Germany

**Joris Frese** orcid.org/0000-0002-5871-997X
European University Institute, Italy

**Oliver Genschow** orcid.org/0000-0001-6322-4392
Leuphana University, Lüneburg, Germany

**Vaitsa Giannouli** orcid.org/0000-0003-2176-8986
School of Medicine, Aristotle University of Thessaloniki, Greece

**Biljana Gjoneska** orcid.org/0000-0003-1200-6672
Macedonian Academy of Sciences and Arts, North Macedonia

**Timo Gnambs** orcid.org/0000-0002-6984-1276
Leibniz Institute for Educational Trajectories, Germany

**Amélie Gourdon-Kanhukamwe** orcid.org/0000-0002-3060-1320
King's College London, United Kingdom

**Christopher J. Graham** orcid.org/0000-0002-1144-7970
Royal College of Physicians of Edinburgh, United Kingdom

**Helena Hartmann** orcid.org/0000-0002-1331-6683
Department of Neurology, Center for Translational and Behavioral Neuroscience, University Hospital Essen, Germany

**Clove Haviva** orcid.org/0000-0003-3266-5755
Dalhousie University, Canada

**Alina Herderich** orcid.org/0000-0002-2940-600X
Graz University of Technology, Austria

**Leon P. Hilbert** orcid.org/0000-0002-4366-9332
Department of Psychology, University of Amsterdam, the Netherlands

**Darías Holgado** orcid.org/0000-0003-3211-8006
Institute of Sport Sciences, University of Lausanne, Switzerland

**Ian Hussey** orcid.org/0000-0001-8906-7559
University of Bern, Switzerland

**Zlatomira G. Ilchovska** orcid.org/0000-0001-6682-9952
School of Psychology, University of Birmingham, United Kingdom; School of Psychology, University of Nottingham, United Kingdom

**Tamara Kalandadze** orcid.org/0000-0003-1061-1131
Østfold University College, Norway

**Veli-Matti Karhulahti** orcid.org/0000-0003-3709-5341
Department of Music, Art and Culture Studies, University of Jyväskylä, Finland

**Leon Kasseckert**
University of Münster, Germany

**Maren Klingelhöfer-Jens** orcid.org/0000-0002-5393-7871
University Medical Center Hamburg-Eppendorf, Germany

**Alina Koppold** orcid.org/0000-0002-3164-3389
University Medical Center Hamburg-Eppendorf, Germany

**Max Korbmacher** orcid.org/0000-0002-8113-2560
Western Norway University of Applied Sciences, Norway

**Louisa Kulke** orcid.org/0000-0002-9696-8619
University of Bremen, Germany

**Niclas Kuper** orcid.org/0000-0001-6901-0205
University of Münster, Germany

**Annalise LaPlume** orcid.org/0000-0001-6725-3270
Toronto Metropolitan University, Canada

**Gavin Leech** orcid.org/0000-0002-9298-1488
Arb Research, United Kingdom

**Feline Lohkamp**
Leuphana University Lüneburg, Germany

**Nigel Mantou Lou** orcid.org/0000-0003-1363-833X
University of Victoria, Canada

**Dermot Lynott** orcid.org/0000-0001-7338-0567
Maynooth University, Ireland

**Maximilian Maier** orcid.org/0000-0002-9873-6096
University College London, United Kingdom

**Maria Meier** orcid.org/0000-0002-1655-5479
University of Konstanz, Germany

**Maria Montefinese** orcid.org/0000-0002-7685-1034
Department of Developmental and Social Psychology, University of Padova, Italy

**David Moreau** orcid.org/0000-0002-1957-1941
University of Auckland, New Zealand

**Kellen Mrkva** orcid.org/0000-0002-6316-5502
Baylor University, Hankamer School of Business, United States

**Monika Nemcova** orcid.org/0009-0002-9941-8716
Charles University, Czech Republic

**Danna Oomen** orcid.org/0000-0003-2638-1975
Leuphana University, Germany

**Julian Packheiser** orcid.org/0000-0001-9805-6755
Department of Social Neuroscience, Ruhr University Bochum, Germany

**Shubham Pandey**
Indian Institute of Technology Bombay, India

**Frank Papenmeier** orcid.org/0000-0001-5566-9658
University of Tübingen, Germany

**Mariola Paruzel-Czachura** orcid.org/0000-0002-8716-9778
University of Silesia in Katowice, Poland; University of Pennsylvania, United States

**Yuri G. Pavlov** orcid.org/0000-0002-3896-5145
University of Tuebingen, Germany

**Zoran Pavlović** orcid.org/0000-0002-9231-5100
University of Belgrade, Serbia

**Charlotte R. Pennington** orcid.org/0000-0002-5259-642X
School of Psychology, Aston University, United Kingdom

**Merle-Marie Pittelkow** orcid.org/0000-0002-7487-7898
QUEST Center for Responsible Research, Berlin Institute of Health at Charité – Universitätsmedizin Berlin, Berlin, Germany

**Willemijn Plomp** orcid.org/0000-0002-3254-5561
Leiden University, the Netherlands

**Paul E. Plonski** orcid.org/0000-0002-6748-6020
Tufts University, United States

**Ekaterina Pronizius** orcid.org/0000-0003-1446-196X
University of Vienna, Austria

**Andrew Adrian Pua** orcid.org/0000-0002-2225-5245
School of Economics, De La Salle University, Manila, Philippines

**Katarzyna Pypno-Blajda** orcid.org/0000-0002-3024-3535
University of Silesia in Katowice, Poland

**Manuel Rausch** orcid.org/0000-0002-5805-5544
Rhine-Waal University of Applied Sciences, Germany

**Tobias R. Rebholz** orcid.org/0000-0001-5436-0253
University of Tübingen, Germany

**Elena Richert** orcid.org/0000-0003-0919-4879
Reykjavik University, Iceland; University of Eastern Finland, Finland

**Jan Philipp Röer** orcid.org/0000-0001-7774-3433
Department of Psychology and Psychotherapy, Witten/Herdecke University, Witten, Germany

**Robert Ross** orcid.org/0000-0001-8711-1675
Macquarie University, Australia

**Kathleen Schmidt** orcid.org/0000-0002-9946-5953
Ashland University, United States

**Aleksandrina Skvortsova** orcid.org/0000-0003-0512-0792
Department of Psychology, Leiden University, the Netherlands

**Matthias F. J. Sperl** orcid.org/0000-0002-5011-0780
Department of Clinical Psychology and Psychotherapy, University of Siegen, Siegen, Germany; Department of Clinical Psychology and Psychotherapy, University of Giessen, Giessen, Germany; Center for Mind, Brain and Behavior, Universities of Marburg and Giessen (Research Campus Central Hessen), Marburg, Germany

**Alvin W. M. Tan** orcid.org/0000-0001-5551-7507
Stanford University, United States

**J. Lukas Thürmer** orcid.org/0000-0002-5315-2847
Paris-Lodron University Salzburg & Private University Seeburg Castle, Austria

**Aleksandra Tołopiło** orcid.org/0000-0002-2518-6759
Center for Research on Biological Basis of Social Behavior, SWPS University, Poland

**Wolf Vanpaemel** orcid.org/0000-0002-5855-3885
KU Leuven, Belgium

**Leigh Ann Vaughn** orcid.org/0000-0002-2399-7400
Ithaca College, United States

**Steven Verheyen** orcid.org/0000-0002-6778-6744
Erasmus University Rotterdam, the Netherlands

**Lukas Wallrich** orcid.org/0000-0003-2121-5177
Birkbeck, University of London, United Kingdom

**Lucia Weber**
Universtät Bamberg, Germany

**Julia K. Wolska** orcid.org/0000-0001-8675-4388
Manchester Metropolitan University, United Kingdom

**Mirela Zaneva** orcid.org/0000-0003-3569-931X
Christ Church College, University of Oxford, United Kingdom

**Yikang Zhang** orcid.org/0000-0001-5173-562X
Faculty of Psychology and Neuroscience, Maastricht University, the Netherlands; Criminology Department, Max Planck Institute for the study of Crime, Security, and Law, Germany

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

## PEER REVIEW COMMENTS

*Journal of Open Psychology Data* has blind peer review, which is unblinded upon article acceptance. The editorial history of this article can be downloaded here:

- **PR File 1.** Peer Review History. DOI: https://doi.org/10.5334/jopd.101.pr1

**TO CITE THIS ARTICLE:**
Röseler, L., Kaiser, L., Doetsch, C., Klett, N., Seida, C., Schütz, A., Aczel, B., Adelina, N., Agostini, V., Alarie, S., Albayrak-Aydemir, N., Aldoh, A., Al-Hoorie, A. H., Azevedo, F., Baker, B. J., Barth, C. L., Beitner, J., Brick, C., Brohmer, H., Chandrashekar, S. P., Chung, K. L., Cockcroft, J. P., Cummins, J., Diveica, V., Dumbalska, T., Efendic, E., Elsherif, M., Evans, T., Feldman, G., Fillon, A., Förster, N., Frese, J., Genschow, O., Giannouli, V., Gjoneska, B., Gnambs, T., Gourdon-Kanhukamwe, A., Graham, C. J., Hartmann, H., Haviva, C., Herderich, A., Hilbert, L. P., Holgado, D., Hussey, I., Ilchovska, Z. G., Kalandadze, T., Karhulahti, V.-M., Kasseckert, L., Klingelhöfer-Jens, M., Koppold, A., Korbmacher, M., Kulke, L., Kuper, N., LaPlume, A., Leech, G., Lohkamp, F., Lou, N. M., Lynott, D., Maier, M., Meier, M., Montefinese, M., Moreau, D., Mrkva, K., Nemcova, M., Oomen, D., Packheiser, J., Pandey, S., Papenmeier, F., Paruzel-Czachura, M., Pavlov, Y. G., Pavlović, Z., Pennington, C. R., Pittelkow, M.-M., Plomp, W., Plonski, P. E., Pronizius, E., Pua, A. A., Pypno-Blajda, K., Rausch, M., Rebholz, T. R., Richert, E., Röer, J. P., Ross, R., Schmidt, K., Skvortsova, A., Sperl, M. F. J., Tan, A. W. M., Thürmer, J. L., Tołopiło, A., Vanpaemel, W., Vaughn, L. A., Verheyen, S., Wallrich, L., Weber, L., Wolska, J. K., Zaneva, M., & Zhang, Y. (2024). The Replication Database: Documenting the Replicability of Psychological Science. *Journal of Open Psychology Data,* 12: 8, pp. 1–23. DOI: https://doi.org/10.5334/jopd.101

**Submitted:** 11 April 2024     **Accepted:** 27 August 2024     **Published:** 11 September 2024

