## [Peer Review History. · Journal of Open Psychology Data]

Reviewer A:

Recommendation: Revisions Required

Comments to the author(s)

Thank you for the opportunity to review this manuscript for the Journal of Open Psychology Data. I commend the authors for their valuable contribution to psychology and replication science through the creation of the Replication Database.

The manuscript offers a clear and thorough description of the database, including the procedures for inputting replication information, validating data, and accessing information from original and replication studies.

While I believe the manuscript is suitable for publication in its current form, I suggest the authors address two minor issues before finalizing it.

Thank you very much for your positive and helpful feedback!

1. Firstly, the quality control procedures for ensuring data accuracy are mentioned but not detailed. A more comprehensive description of the validation process would be beneficial. Specifically, what steps are taken when there is a discrepancy between original and replication studies? Are the authors who input the data notified when changes are made due to inconsistencies or other reasons? Additionally, are there circumstances under which a second person may be involved in the validation process?

I translated the specific instructions, uploaded them to the project, and added the link in the quality assurance section (<https://osf.io/x45js>).

We added information about dealing with inconsistencies. Here is the link to an error that a research assistant and co-author spotted in a paper:

<https://pubpeer.com/publications/7613DC7756A1F876DCECA642AEA2E2>.

Dealing with Inconsistencies

In cases of inconsistencies, we corrected values to match the source material. We identified an error in a replication report, confirmed it with the author(s), and commented on PubPeer. If authors were unreachable, we relied on the original or replication reports. After other researchers flagged two errors in the CurateScience data (LeBel et al., 2018), we revalidated all CurateScience entries by comparing effect and sample sizes directly with the original reports rather than the database. For future errors in our database, we encourage researchers to submit a comment to this article via PubPeer (<https://pubpeer.com>).

2. Secondly, the authors highlight the database's value for teaching and learning on page

3 but only briefly elaborate on this in the final bullet point of section 4, Reuse Potential. I would appreciate it if the authors could expand on the potential educational uses of the database and whether there are plans to develop additional resources to support education in replicability and replication studies.

We added two more things to reuse potential – teaching.

- **Teaching:** Textbooks and teaching materials are highly likely to include findings that could not be replicated but past problems in findability made it difficult to provide a more nuanced discussion. The Replication Database can help researchers revising these materials and including more recent findings for the discussed phenomena or theories via a reference list annotation tool. This way, literature list can be read and annotated with respect to replication attempts (e.g., if there have been any replication attempts and what their outcomes were). Moreover, instead of relying on singular findings, teachers can for example ask students to take a look at replications, compare them with the original findings, and thereby help them develop skills to evaluate bodies of research. Finally, replication studies have become an integral part of undergraduate research (Boyce et al., 2023; Jekel et al., 2020; Korell et al., 2023; Quintana, 2021). The database provides a low-threshold opportunity to make student replications visible.

Apart from that, we are working on multiple projects. That will (hopefully) support teaching. As all of these are work in progress, we do not mention them in the manuscript.

Thank you again for the opportunity to review this important work.

Reviewer B:

Recommendation: Accept Submission

Comments to the author(s)

Thank you for the opportunity to read this interesting manuscript! In the present data publication, the authors introduce their database of replication studies in psychology. I fully agree with the authors that the replication crisis is one of the most pressing issues psychological science currently faces. Thus, I highly commend their efforts in creating this database and think that it addresses a highly relevant need within the scientific

community.

The manuscript is of high quality and thoroughly describes the dataset and its structure. The authors also provide additional online resources, freely accessible through OSF (Open Science Framework). These resources make the process of the creation of the database very transparent. One minor consideration is the long-term persistence of the links to these OSF resources. Currently, all links are working, but reading through the manuscript I wondered whether some of these files might be moved or deleted/replaced later.

In summary, the manuscript presents a valuable and well-structured database that addresses a crucial issue in psychological research and I therefore find it very suited for publication in JOPD. In the following, I will comment on some minor needs for clarification. Addressing these comments could further strengthen the manuscript.

Thank you very much for your positive and helpful feedback!

Section 2.6

“As a collaborative community effort from the contributors, all mandatory fields (Table 2) were systematically verified by one person per entry. [...]”

I would like to have more information on who exactly these persons were. How many persons were involved in this process? Were these persons trained or provided with specific guidelines?

We updated the section on the validators:

As a collaborative community effort from the contributors, all mandatory fields (Table 2) were systematically verified by one person per entry (listed in the variable `validated_person`). These seven contributors were students fulfilling course credits or research assistants. They were acquainted with statistical methods (e.g., effect sizes and null hypothesis significance testing) and used standardized instructions (<https://osf.io/y3fm8>). For example, they tested hyperlinks, and assessed descriptions and keywords for plausibility. The attribution of effects to one or multiple samples and the accurate naming of Sample IDs were also examined.

“Afterwards, one of two project team members checked whether the entries regarding effect sizes, sample sizes, and references in the ReD Database aligned with those of the submitted studies.”

Once more, who are these two project team members and why is it important to mention that there were two project team members if only one of them did anything in the data validation process? (if I understand it correctly)

We apologize for the confusion. We revised the paragraph and clarified it:

For each batch of submissions, a project team member checked whether the entries regarding effect sizes, sample sizes, and references in the Replication Database aligned with those of the submitted studies. These were again research assistants or the project lead. In some cases, the original authors of large datasets validated the entries or converted the data.

(3) Dataset Description and Access

Readers might be interested in how they should cite data usage. Do you want them to cite the OSF-project page and the dataset version? Or the JOPD article? I would recommend to offer some sort of guidance on this issue in the article. (I am aware that there is some information on this issue on the project’s website. However, this information should also be part of the data publication in JOPD.)

Thank you for this suggestion! We agree and added a recommended citation to the section 3.7 Limits to Sharing.

Recommended citation: Please cite this article and along with it the most recent version of the OSF-project (<https://osf.io/9r62x/>) that includes a version number and contributors who joined the project since 04/2024.

Limitations

One final thought: As with all crowd science projects, the long-term development of the replication database project is uncertain. The article could emphasize more strongly which parts of the project outputs are assigned DOIs in repositories and will therefore be available in the long term (e.g., ten years from now) and for which parts this is less certain. For example, the shinyapp really looks great and is very useful as a starting point but I am not sure whether the underlying code will run in a few years without proper maintenance. Please note that this is no “real” limitation with regard to this article because the dataset that the project has generated and that the article presents is of invaluable worth to the research community in its own right. Other readers might, however, have the same thoughts (or read this article in ten years), so it might be worthwhile to mention this aspect.

We created a frozen version by registering the entire project in its current state. We replaced all links with the permanent links from the frozen version. We also highlighted all non-permanent links. Here is an example of the Dataset Description and Access section.

The datasets and materials are openly available in the OSF repository (<https://osf.io/9r62x/>) and will be updated continuously as the database grows.

- Dataset used for the reported analyses: <https://osf.io/qtkzy>
- Google Sheets spreadsheet that we plan to update regularly: https://docs.google.com/spreadsheets/d/1x68oW2H_Xrdv44fleycl4fegsmQgCa60GxeZZ_hAR90/edit?pli=1#gid=1463805480 (non-permanent link)
- Interactive Shiny Application: <https://metaanalyses.shinyapps.io/replicationdatabase/> (non-permanent link)
- Dataset changelog (starting January 2024): <https://osf.io/ej46t>

Figure 2

There is a typo “Janes” vs. “Failure to replicate James” in ID6.

Thank you for pointing this out, we corrected it to say “Janes” in all instances.